# Host–Pathogen Interactions in Leaf Petioles of Common Ash and Manchurian Ash Infected with *Hymenoscyphus fraxineus*

**DOI:** 10.3390/microorganisms10020375

**Published:** 2022-02-05

**Authors:** Lene R. Nielsen, Nina E. Nagy, Sara Piqueras, Chatchai Kosawang, Lisbeth G. Thygesen, Ari M. Hietala

**Affiliations:** 1Department of Geosciences and Natural Resource Management, University of Copenhagen, 1958 Frederiksberg, Denmark; sps@ign.ku.dk (S.P.); chko@ign.ku.dk (C.K.); lgt@ign.ku.dk (L.G.T.); 2Division of Biotechnology and Plant Health, Norwegian Institute of Bioeconomy Research (NIBIO), 1431 Ås, Norway; nina.elisabeth.nagy@nibio.no; 3Norwegian Institute of Bioeconomy Research (NIBIO), 7734 Steinkjer, Norway; ari.hietala@nibio.no

**Keywords:** ash dieback, *Fraxinus excelsior*, light microscopy, plant-pathogen interaction, real-time PCR

## Abstract

Some common ash trees (*Fraxinus excelsior*) show tolerance towards shoot dieback caused by the invasive ascomycete *Hymenoscyphus fraxineus*. Leaf petioles are considered to serve as a pathogen colonization route to the shoots. We compared four common ash clones with variation in disease tolerance, and included the native host, Manchurian ash (*Fraxinus mandshurica*), as a reference. Tissue colonization, following rachis inoculation by *H. fraxineus*, was monitored by histochemical observations and a quantitative polymerase chain reaction (qPCR) assay specific to *H. fraxineus*. Axial spread of the pathogen towards the petiole base occurred primarily within the phloem and parenchyma, tissues rich in starch in healthy petioles. In inoculated petioles, a high content of phenolics surrounded the hyphae, presumably a host defense response. There was a relationship between field performance and susceptibility to leaf infection in three of the four studied common ash clones, i.e., good field performance was associated with a low petiole colonization level and vice versa. Low susceptibility to leaf infection may counteract leaf-to-shoot spread of the pathogen in common ash, but the limited number of clones studied warrants caution and a larger study. The Manchurian ash clone had the highest petiole colonization level, which may suggest that this native host has evolved additional mechanisms to avoid shoot infection.

## 1. Introduction

Since the 1990s, common ash forests have become increasingly damaged because of the rapid spread of an invasive ascomycete, which causes shoot dieback and mortality of ash species native to Europe. The causative fungus, *Hymenoscyphus fraxineus* Baral, Queloz, Hosoya (syn. *H. pseudoalbidus*) [1], originates from Asia, where it appears to be a harmless associate of local ash species, *Fraxinus mandshurica* Rupr. and *F. chinensis* Roxb. subsp. *rhynchophylla* (Hance) E. Murray (see [2] and references therein). In Europe, the fungus acts as a severe pathogen on *Fraxinus excelsior* L. (common ash), but also the other European *Fraxinus* species are affected, with *F. angustifolia* Vahl (narrow-leaved ash) being highly susceptible and *F. ornus* L. (manna-ash or flowering ash) only weakly susceptible [3,4,5].

Host infection takes place when the airborne ascospores of *H. fraxineus* germinate on the leaf surface and infect the foliage, followed by hyphal growth through the leaf petiole to shoots prior to autumn leaf shed [6]. Besides the leaf-to-shoot route, lenticels [7] and direct penetration of the epidermis of young shoots [8] may offer the pathogen additional entrances to the stem tissues. Typical disease symptoms are necrotic lesions on shoots. The infected shoots die once the lesion-associated cambial death has extended all the way around the shoot circumference, and the progressive crown reduction eventually results in the death of the tree. In European arboreta, the pathogen seems to effectively infect leaf tissues of several native Asian host species, as well as the European ones, the difference being that no or only minor shoot symptoms are induced in Asian trees [5]. Upon artificial stem inoculation, the Asian host species show short lesion lengths in comparison to common ash [5].

Studies from field trials and seed orchards have revealed that the level of shoot dieback varies highly among common ash clones, and that a minority of clones develop very few symptoms [9]. The field performance of half-sib families of common ash has further revealed that genetic differences explain a high proportion of the variation in resistance to ash dieback (e.g., [9]). Different approaches have been used to identify the genetic mechanisms behind ash dieback resistance [10,11,12]. The most comprehensive of the listed studies used a Genome-Wide Association Study approach, and identified 192 highly significant single-nucleotide polymorphisms (SNPs) associated with ash dieback resistance; 61 SNPs were located either within or in close proximity to genes that have been reported to participate in host defense in other plant species [12]. The remaining 131 SNPs associated with resistance to ash dieback were not coupled with any genes known to regulate plant–pathogen interactions in other pathosystems.

As the petiole-to-shoot route is considered to be the main path for shoot infection by *H. fraxineus*, we hypothesized that the reduced spread of the pathogen and more pronounced defense responses would take place in the leaf tissues of common ash genotypes with a strong field performance, i.e., trees with a low level of shoot dieback in field locations with an epidemic level of the disease. We tested our hypothesis using a *H. fraxineus*-specific real-time PCR assay to monitor the progress of *H. fraxineus* colonization and by registering cytological host defense responses in leaf petioles of *F. excelsior* clones, with variation in susceptibility to shoot dieback and with the native host Manchurian ash serving as a reference.

## 2. Materials and Methods

### 2.1. Plant Materials

Grafted potted plants of ash genotypes, displaying variation in susceptibility to shoot dieback when grown under field conditions, were kept in a greenhouse free of *H. fraxineus* during the experiment. The common ash genotypes originated from two trials, the clonal trial “Tapsøre” (FP281) established in 1998 and located at 55°24′14.2″ N 9°27′27.5″ E [13], and the progeny trial “Randers” (F384) established in 2004 and located at 56°30′00.0″ N 10°02′24.0″ E [14]. The clones varied in susceptibility level, measured as the average percent crown damage score across five years of measurements of the individual trees in the trials. Clones 35 and 14165 had few crown symptoms, and showed a crown damage score of 5% and 8% across five years of measurements, respectively. Clones 40 and 27x showed a crown damage score of 34% and 18%, respectively. Regarding clone 27x, the ramet of the susceptible clone 27 (coordinate position 12;26 in the trial) previously used for the ash dieback experiments of our research group had died in the trial, and therefore a new ramet was used (coordinate position 01;17). Later genotyping revealed that the grafted plant with position 01;17 had a deviating genotype from the original clone 27, and was therefore denoted clone 27x. For the native host, Manchurian ash clone 1989-0095-4 from the Hørsholm arboretum (55°52′01.9″ N 12°30′23.3″ E) was included in the study. This clone did not show crown symptoms when surveyed in a previous study [5]. See Appendix A for more information about the plant material and average damage scores.

### 2.2. Inoculation Experiments

Inoculations with ash wood plugs (0.5 cm in length and 0.2 cm in diameter) colonized by the virulent *H. fraxineus* strain 18.3 [15] were performed on 6–7 June 2018. The inoculum was cultured on malt extract agar with sterilized wood plugs for three to four weeks. Three ramets of each clone were inoculated on two or three compound leaves. In each case, the rachis was wound-inoculated between the first and second leaflet pair, counted from the petiole base (Figure 1a,b), by cutting a 1-cm-long superficial incision in the rachis. Controls were inoculated similarly with sterile wood plugs (two controls per ramet). Leaves that dropped off before the end of the experiment were, in most cases, found and collected and stored in the freezer. On 24 September, we terminated the experiment, collected all the remaining leaves, and stored them in the freezer until further processing. The number of collected and analyzed inoculated rachises were six for clone 35, seven for clone 14165, seven for clone 40, five for clone 27x, and three for Manchurian ash clone 1989-0095-4. In the case of Manchurian ash, only rachises from two of the three inoculated ramets were recovered.

We measured the length of the visible necroses from the center of the incision towards the petiole base (Figure 1c), and also estimated the extension of necrosis around the rachis at the inoculation point, referred to as the necrosis circumference in later text (a value of 100% indicates that necrosis continued all around the rachis). The following four approximately 5-mm-long sample types (S1–S4) were collected: S1, at the border of the necrosis; S2, 1 cm away from the first sample towards the petiole base; S3, petiole base; and S4, petiole–twig junction (Figure 1a). In one case, an additional sample was taken 1 cm away from S2 towards the petiole base. For some of the leaves that dropped off before 24 September, we could not collect all four sample types. The samples were divided longitudinally into two halves, one reserved for DNA extraction and the other for microscopy. The sample for microscopy was stored in fixative containing 3 mL of 2% paraformaldehyde and 1.25% glutaraldehyde in 50 mM L-piperazine-N-N′-bis (2-ethane sulfonic) acid buffer (pH 7.2) at 4 °C for subsequent processing and light microscopy examination, as described below. Samples meant for DNA extraction were weighed to allow for normalization of the pathogen (*H. fraxineus*) DNA amount in relation to the tissue weight. Typical symptoms appearing after rachis inoculation are shown in Figure 1c.

### 2.3. DNA Extraction and Real-Time PCR Analysis

Samples were collected, as shown in Figure 1a, and DNA extraction was performed with the E.Z.N.A.^®^ SP Plant DNA Kit (Omega Bio-tek, Norcross, GA, USA) following the manufacturer’s protocol and by eluting with 30 µL elution buffer.

For real-time PCR analysis, a ten-fold dilution series (undiluted, 10× and 100× diluted) was prepared for each DNA sample, and duplicates of each dilution were run on an Mx3005P qPCR system (Agilent Technologies, Santa Clara, CA, USA) using the *H. fraxineus* DNA specific primer/probe set described previously [16], and PrimeTime Gene Expression Master Mix (Integrated DNA Technologies, Leuven, Belgium). The PCR program was composed of 3 min at 95 °C, followed by 40 cycles of 95 °C for 15 s, and 60 °C for 60 s. The MxPro qPCR software (Agilent Technologies) was used to determine the cycle thresholds (Ct values). The standard curve for *H. fraxineus*, generated from a six-fold dilution series of pure *H. fraxineus* DNA, was based on the relationship of known DNA concentrations (y) and the corresponding average Ct value (x) from four PCR replicates per DNA concentration: log y = −0.2746x + 8.3152 (*R*^2^ = 0.9991). The average Ct value of the two 10× dilutions of the experimental templates was used in calculations to avoid the effects of PCR inhibitors in the undiluted sample.

### 2.4. Data Analysis of Lesion Size and H. fraxineus DNA Content

To test if there were clonal differences following rachis inoculation, we compared the lesion length, extent of necrosis circumference at the point of rachis inoculation, and amount of *H. fraxineus* DNA among the four common ash clones (35, 14165, 40, and 27x). Differences among clones were tested based on F-tests in a general linear model, assuming independent and normal distributed residuals. We also tested for differences between the two more susceptible clones and the two less susceptible clones; here, differences were tested against the variation among clones within the health groups (i.e., based on a model that assumed clones within the health group as a random effect).

For *H. fraxineus* DNA, the data were pooled for each clone (samples S1 to S4 = total DNA) before statistical analysis. Before performing F-tests, we applied square root transformation for necrosis circumference and logarithmic transformation for lesion length and DNA amount to fit the model assumptions. Correlation analyses were applied to quantify the relationships between necrotic lesion lengths and circumference, and the amount of *H. fraxineus* DNA in rachis samples. Statistical analyses were made in SAS v. 9.4 [17] with the model test fit based on visual inspection of the residuals.

### 2.5. Light Microscopy and Staining

The tissue pieces were processed and embedded in L. R. White resin (TAAB, Berks, UK), and were stained as previously described [18]. For infected rachis samples, only complete series with *H. fraxineus* DNA detected in all four sample types (samples S1 to S4) were examined (common ash clones 35, 14165, 27x, and Manchurian ash clone 1989-0095-4). Semi-thin cross sections (1.5 μm) were cut using an LKB 2128 Ultratome (Leica Microsystems, Wetzlar, Germany), and were dried onto glass slides. Sections used for routine observations were stained with Stevenel’s blue. Periodic acid-Schiff (PAS; Sigma-Aldrich, St. Louis, MO, USA) staining was used to detect starch grains and carbohydrate-rich compounds (see [18]). Unstained sections were examined with autofluorescence (natural emission) for the detection of polyphenolic and suberin components using a Leica DMR light microscope operated in epifluorescence mode. Blue light (450–490 nm) and a long-band-pass filter (>520 nm) was used for the excitation and visualization of polyphenolics, while ultraviolet light was used for the excitation of suberin that mainly fluoresces in the blue region. Nile red was used to stain neutral lipids [19], which were subsequently detected as yellow–gold droplets when illuminated with UV-light (330–380 nm).

To estimate the prevalence of starch and fungal hyphae within the phloem, axial parenchyma, and perimedullary parenchyma, their presence was recorded in ten randomly chosen microscopic fields with 100× objective lens and a microscopic field area of 0.0314 mm^2^.

## 3. Results

### 3.1. Lesion Size and H. fraxineus DNA Level Profiles in Inoculated Rachises

A brown necrotic tissue zone appeared around the site of inoculation (Figure 1c), and after 16 weeks of incubation, the necrotic tissue extended 4 to 38 mm towards the petiole base from the site of inoculation (Figure 2a and Table 1).

The two common ash clones 27x and 40 with an inferior field performance (average PDS of 18% and 34%, respectively) had the longest necrotic lesions, followed by the Manchurian ash clone and the two less susceptible common ash clones 35 and 14165 (PDS of 5% and 8%, respectively) (Figure 2a). The relative necrosis circumference was also slightly more pronounced in the susceptible common ash clones 27x and 40, by 96% and 79%, respectively, compared to 77% and 65% for the less susceptible common ash clones 35 and 14165, respectively (Figure 2b). Despite the small dataset, the differences in lesion lengths were close to significant both among clones (F = 2.86, *p* = 0.062) and between the two groups of susceptible and less susceptible clones (F = 9.71, *p* = 0.089), while there were no significant differences among clones in the necrosis circumference or pathogen DNA levels.

The real-time PCR-based spatial DNA level profiles of *H. fraxineus* in rachises 16 weeks after inoculation are shown in Table 1. Irrespective of clone, considerably higher amounts of pathogen DNA were detected from the samples taken at the lesion border or 1 cm away from it in comparison to samples from the petiole base or twig junction (Figure 2c, Table 1). In clones 35, 14165, and 40, very low amounts of the pathogen DNA were generally detected (Figure 2d). Comparably high levels of pathogen DNA were measured in the clone 27x, while by far the highest amount of pathogen DNA was found in the Manchurian ash clone 1989-0095-4, where the amount detected at the lesion border was almost six-fold higher compared with the corresponding sample from clone 27x. Pathogen DNA was not detected in any of the analyzed controls (inoculated with sterile wood plugs). For common ash, the amount of pathogen DNA at the lesion border (S1) was significantly correlated with the amount detected 1 cm from the lesion border (S2), and the amount registered 1 cm from the lesion border (S2) was in turn significantly correlated with the amount detected at the petiole base (S3) (*p* = 0.042 and *p* < 0.001, respectively). Samples S3 and S4 (petiole–twig junction) showed no correlation in pathogen DNA amount (*p* = 0.086). The respective correlations between positions S1 and S3 (*p* = 0.066), and between S1 and S4 (*p* = 0.331) were not significant.

No significant correlation was found between lesion length and necrosis circumference, and the amounts of DNA in samples S1–S4 were not correlated with necrosis circumference or lesion lengths. The latter is also evident in Table 1, which shows that in many cases, long lesions were not accompanied by the detection of *H. fraxineus* DNA in tissue samples far from the inoculation site (S3 and S4).

### 3.2. Hyphal Colonization and Host Defence Responses

The following cell types were identified in the petioles (mentioned in the order from surface towards petiole center), epidermis, collenchyma, cortex, phloem, xylem, axial parenchyma, perimedullary parenchyma, and pith (Figure 3a). However, axial and perimedullary parenchyma were not observed in the samples from the petiole base and leaf/twig junction. Concerning specific cell types, fungal hyphae with an average diameter of 4 μm showed a clear affinity to phloem and axial and perimedullar parenchyma in the infected petioles (Figure 3b), whereas in uninfected tissues, these cells were devoid of fungal propagules and were rich in starch (Table 2).

Microscopic observations of the frequency of fungal propagules in sections made from inoculated petioles were in agreement with the corresponding qPCR estimates for *H. fraxineus* DNA quantity (Table 2). Sections from the Manchurian ash clone 1989-0095-4 and the common ash clone 27x, with high *H. fraxineus* DNA levels at the lesion border and 1–2 cm from it, showed abundant fungal propagules in these regions: most of the ten examined microscopic fields from phloem, axial parenchyma, and perimedullary parenchyma showed the presence of fungal propagules, but generally lacked starch (Table 2). Samples from the common ash clones 35 and 14165, with low *H. fraxineus* DNA levels at the lesion border and 1–2 cm from it, showed few fungal propagules: most of the ten microscopic fields from phloem, axial parenchyma, and perimedullary parenchyma showed no fungal propagules, but the presence of starch. Tylosis-like structures were occasionally observed in the vessels of infected plants (Figure 3c,d).

Hyphae were growing mostly axially in the parenchyma (Figure 3c), but also indications of tangential spread between neighboring cells were observed—the latter was observed in the perimedullary pith (Figure 3d), phloem, and inner cortex parenchyma. The parenchyma cells hosting fungal propagules had generally little or no starch, and the hyphae were commonly surrounded or filled with phenolic material. While at the lesion border, hyphae were also relatively frequent in the cortex collenchyma and parenchyma, 10 mm away from the lesion edge, hyphae were primarily present in the axial xylem parenchyma and the marginal perimedullary parenchyma next to the spongy pith (Figure 3e,f). Of the inoculated clones, fungal colonization of inner cortex parenchyma was particularly pronounced in the Manchurian ash clone 1989-0095-4, and this clone was also the only one where fungal propagules, although at a low level, were observed at the petiole–twig junction.

As shown in Figure 4, UV and blue light autofluorescence revealed the presence of both suberin (Figure 4a,b) and polyphenols (Figure 4c,d), which was especially pronounced in the infected tissues. Polyphenols were especially pronounced in the phloem and perimedullary parenchyma, taking both grainy deposit appearances (Figure 4c), as well as more homologue aggregates (Figure 4d) in the rows of axial parenchyma aligning the vessels at the lesion border or 1 cm away from it. Fungal hyphae were also present in these cells, often surrounded by polyphenolics (Figure 4d,e). Lipid substances were also observed, owing to positive staining with Nile red (Figure 4f). The inoculated clones of common ash and Manchurian ash showed no obvious difference in the occurrence of phenolics. 

## 4. Discussion

### 4.1. Fungal Colonization and Host Responses in Leaf Rachis and Petiole Tissues

The real-time PCR data from the four common ash clones and one Manchurian ash clone employed in the rachis inoculation experiment divided the clones into two groups: one with high levels of *H. fraxineus* DNA around the site of inoculation and even at the petiole base, and the other with low pathogen DNA levels at the sampled rachis and petiole regions. There was strong agreement between the real-time PCR-based profile of *H. fraxineus* DNA and the prevalence of hyphae in the four clones examined with light microscopy (common ash clones 35, 14165, 27x, and Manchurian ash clone 1989-0095-4). Among these, clones 27x and 1989-0095-4, with abundant hyphal colonization, showed the highest levels of pathogen DNA. The hyphae of *H. fraxineus* have no specific morphological characteristics that allow for reliable in situ identification, but the rather uniform size of the observed hyphae and the consistency between cytological observations and qPCR data suggest that *H. fraxineus* was the primary colonizer of the inoculated tissues.

The highest DNA level of *H. fraxineus* in the inoculated rachises was detected around the site of inoculation, with a rapidly declining level towards the petiole base. There was no correlation between the longitudinal or circumferential extension of the cortex lesion induced by inoculation in rachis, and the distance that the pathogen had advanced towards petiole base. Supported by microscopy, cortex colonization by *H. fraxineus* mycelia was restricted to rachis areas immediately adjacent to the site of infection, whereas in the more distal rachis and petiole regions, fungal hyphae were present primarily within the vascular cylinder. The advancement of hyphae far beyond the cortex lesion within the vascular cylinder has also been observed in naturally infected shoots of common ash [20]. The spread pathway is consistent with a previous study [21], where the same inoculation approach was used as in this experiment. Those authors found no correlation between the extension of cortex lesion in the rachis/petiole tissue and the appearance of symptoms in distal leaflets or the timing of leaf abscission.

In the present work, hyphae showed a high affinity to parenchyma cells within the vascular cylinder, and similar hyphal trailing of the starch pathway was also observed in naturally infected shoots of common ash [20]. While these parenchyma cells showed variable levels of starch in healthy control tissues, starch was sparse in the parenchyma colonized by fungal hyphae, and spherical fluorescing droplets were a common feature of parenchyma cells of the colonized vascular cylinder. This can be attributed to cellular reprogramming by the host to convert the energy reserves to defense compounds, coupled with the eventual fungal assimilation of starch, with phenolic parenchyma deposits being observed only in the inoculated rachis/petiole tissues.

The composition and effect of phenolics induced in ash leaf tissues of different clones upon *H. fraxineus* infection remain to be established. Constitutive foliar phenolics of ash species include coumarins, phenolic acids, and various flavonoids [22], and certain coumarins (fraxetin and esculetin) have been shown to be associated with ash dieback susceptibility [23]. Interestingly, common ash clones differing in susceptibility to shoot dieback showed consistent differences in constitutive twig bark phenolics, but not in constitutive leaf tissue phenolics [24]. Quantitative chemical analyses of the pathogen induced metabolites in rachis/petiole tissues are needed to consider whether clones with different levels of susceptibility to leaf infection show consistent differences in the type and amount of defense compounds.

Finally, it is noteworthy that tylosis-like structures were occasionally formed in the vessels of infected petioles. These were obviously formed by the axial parenchyma aligning the vessels. The low frequency of tyloses and the observation that vessels were normally not blocked by tyloses made it difficult to conclude whether they have any direct effect on the pathogen, especially when considering the finding that axial spread of the hyphae towards the petiole does not take place in the vessels but in the parenchyma.

### 4.2. Susceptibility to Leaf Infection and Shoot Dieback

The common ash clones 35 and 14165 with low levels of shoot dieback in the field conditions showed lower levels of pathogen colonization in inoculated rachises/petioles than the common ash clone 27x, with a poor field performance, and the Manchurian ash clone. The common ash clone 40 with a poor field performance did not fit this pattern, as it showed pathogen colonization levels that were comparable to clones 35 and 14165.

Clone 35 was one of the two clones of common ash that were exposed to spore inoculation in a previous study [5], the other one showing higher susceptibility to ash dieback under field conditions. Similar to the present study, pathogen colonization (verified by real-time PCR) was less prominent in the leaflets of clone 35 than in the leaflets of the more susceptible common ash clone.

The high petiole colonization rate of the Manchurian ash clone by *H. fraxineus* observed here is coherent with the observed abundant ascomata production of the pathogen on overwintered rachises and petioles of this clone (1989-0095-4) in a Danish arboretum [5]. However, this specific Manchurian ash clone showed no shoot dieback in the arboretum, where several other ash species were severely affected by shoot dieback [5]. Our data, in combination with [5], could suggest that low susceptibility to leaf infection counteracts the leaf-to-shoot spread of the pathogen in common ash clones, while Manchurian ash has evolved a mechanism to block this pathway [25,26]. The latter conclusion is consistent with field data that show that in the native range, *H. fraxineus* is restricted to the leaves of Manchurian ash [2,25,26]. However, the limited number of clones included in the present study and the mismatch between field performance and petiole colonization by the pathogen for the common ash clone 40 warrants caution. The host–pathogen interactions that underlie the different levels of susceptibility of common ash clones to leaf colonization by *H. fraxineus* remains to be established.

When considering the mechanisms that can prevent the leaf-to-shoot spread of the pathogen, timing of leaf senescence and shed has been proposed to affect the susceptibility of an ash tree to shoot infection, with common ash clones with early senescence showing less shoot symptoms than the clones with late senescence [13]. In the case of Manchurian ash trees inoculated in the current study, one possible explanation for the early leaf shed could be that the plants responded to *H. fraxineus* infection by premature completion of the abscission layer at the petiole base in order to avoid shoot compromise. Such a scenario would involve a series of physicochemical changes at the petiole base, but our sampling scheme did not allow us to consider this possibility. Whether an induced leaf senescence could hinder mycelial petiole-to-shoot spread might depend on the rapidity of the process, which has been reported to take place over several weeks in September upon natural autumn senescence of healthy white ash (*Fraxinus americana* L.) [27]. However, the correlation between field performance and necrosis development after stem inoculation of common ash clones suggests that the pathogen can also be constrained by the defense mechanisms present in the shoot tissues [28].

### 4.3. Follow-Up Work

As documented in the present study, *H. fraxineus* spreads within the vascular tissues of leaves towards the shoot. This spread takes place primarily within the parenchyma cells, which can convert energy reserves to defense compounds and can also function as defense signal transduction pathways. We cannot reject the hypothesis that there is a connection between leaf and shoot susceptibility, as reduced spread of the pathogen was found in clones with relatively good field performance, i.e., little crown damage. One of the common ash clones deviated from this pattern—a larger dataset is required for clarification. We found, however, no relation between degree of petiole colonization and histochemical features related to defense responses.

It can be envisaged that the timing of the formation and degree of completion of a protective cork layer under the petiole abscission zone, coordinated by parenchyma cells that reside on both sides of the abscission zone [29], may be crucial for hindering the leaf-to-shoot spread by *H. fraxineus* or direct infection of leaf scars after leaf abscission. Studies on whether clones of common ash or other ash species show constitutive or pathogen-induced differences in the chemistry or completion of this protective cork layer are warranted.

## Figures and Tables

**Figure 1 microorganisms-10-00375-f001:**
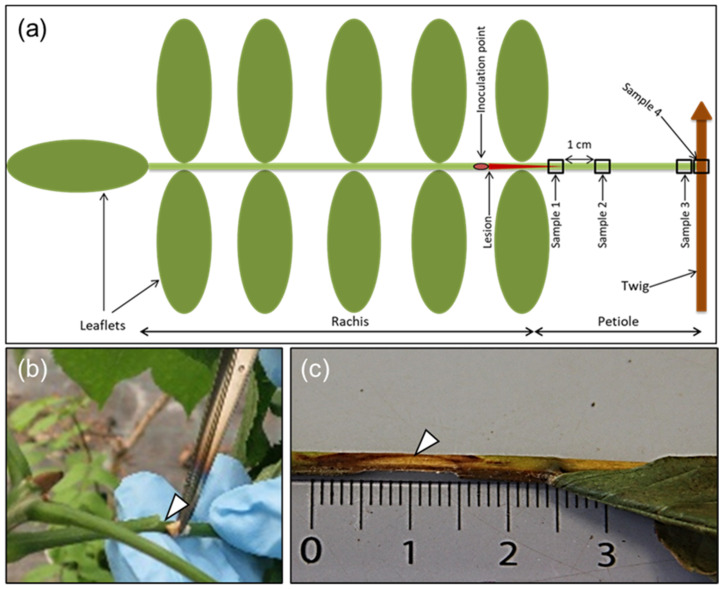
Illustration of rachis sample types, rachis inoculation, and necrosis development in ash. (**a**) A diagram indicating the rachis inoculation site and petiole sampling positions (shown as numbered frames): sample 1 (referred to as S1 in the text and tables), at the border of the lesion enclosing parts of the visible necrosis; sample 2 (S2), 1 cm from border of lesion; sample 3 (S3), base of petiole; sample 4 (S4), junction between petiole and twig. (**b**) Procedure for the inoculation of rachises with an ash wood plug colonized by the pathogen. (**c**) A characteristic necrotic lesion showing an extension from the inoculation site, 16 weeks after inoculation.

**Figure 2 microorganisms-10-00375-f002:**
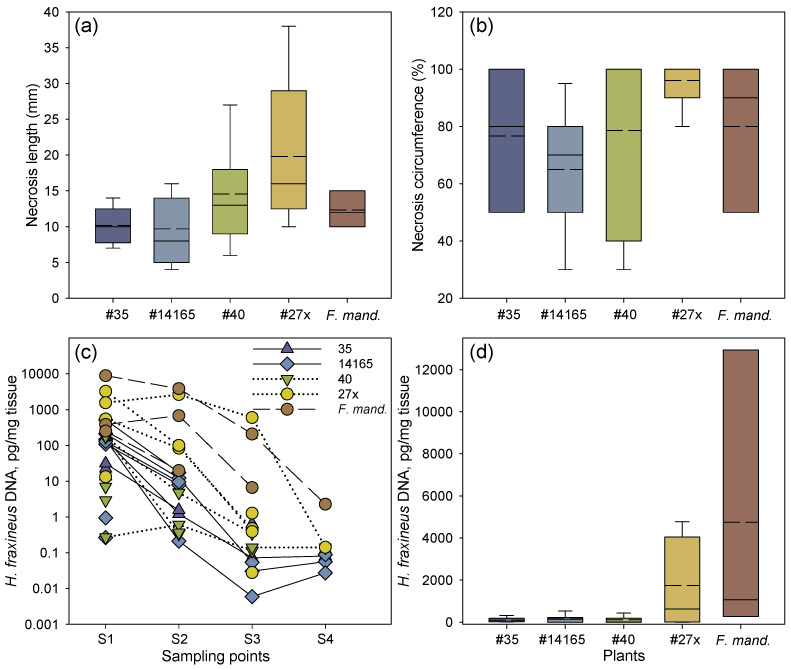
Necrosis development and amount of *H. fraxineus* DNA after rachis inoculation. (**a**) Proximal length of necrosis from the inoculation site. (**b**) Relative necrosis circumference (%) at the inoculation site. The median values for clone 40 and 27x coincide with a necrosis circumference of 100%. (**c**) *H. fraxineus* DNA at each of the four sampling points in inoculated rachises of common ash clones less susceptible to shoot dieback (35 and 14165) or susceptible to shoot dieback (27x and 40), and one clone of Manchurian ash (*F. mand.*, clone 1989-0059-4). Samples connected by a line represent the same inoculated rachis. Samples not connected by a line represent petioles, where *H. fraxineus* DNA was not detected in all of the sampling points. Sampling points are as follows: S1, at the border of the necrosis; S2, 1 cm away from the first sample towards the petiole base; S3, petiole base; and S4, petiole–twig junction. (**d**) Overall mean of *H. fraxineus* DNA per rachis across the four sampling points in each common ash and Manchurian ash clone. Samples size: 35, *n* = 6; 40, *n* = 7; 14165, *n* = 7; 27x, *n* = 5; 1989-0095-4, *n* = 3. For box plots (**a,b,d**), mean (dashed lines) and median values (solid lines) are shown, as well as whiskers representing the minimum and maximum values, and boxes for the 25% and 75% percentiles.

**Figure 3 microorganisms-10-00375-f003:**
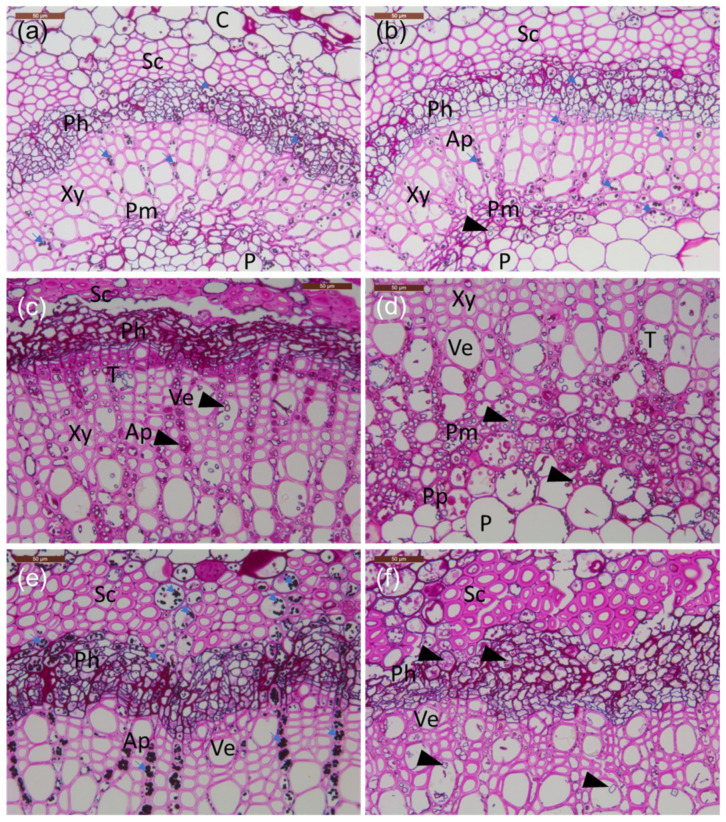
Transverse sections of petioles from the lesion border of inoculated rachis samples compared to healthy tissue. (**a**–**e**) Common ash clones; (**f**) Manchurian ash clone. (**a**) Healthy clone 35. (**b**) Infected clone 35. (**c**) Infected clone 27x; note the phloem with less amount of starch and presence of large amounts of hyphae and phenolics. (**d**) Infected clone 27x; note the abundant presence of mostly axially orientated hyphae in the perimedullary pith. (**e**) Infected clone 14165 with a large amount of starch grains filling the cortex parenchyma, phloem, and axial parenchyma aligning vessels. (**f**) Infected Manchurian ash clone 1989-0095-4 with a low amount of starch and the presence of axially and occasionally laterally orientated hyphae. All tissue sections were stained with PAS carbohydrate specific staining. C: cortex; Ph: phloem; Xy: xylem; P: pith; Sc: sclerenchyma; Pm: perimedullary parenchyma; Pp: polyphenolics; T: tylosis; Ap: axial parenchyma; Ve: vessel elements; small blue arrows: starch grains; arrowheads: fungal hyphae. Scale bars: (**a**–**f**), 50 µm.

**Figure 4 microorganisms-10-00375-f004:**
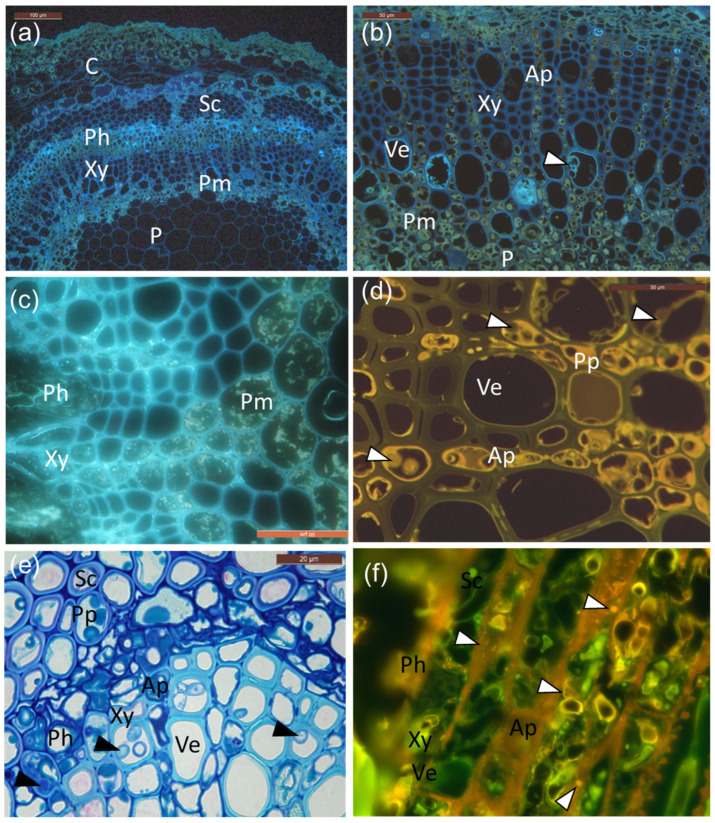
Cellular localization of suberin and phenolics in infected petioles. (**a**) UV auto-fluorescence of infected petioles showing suberin as bright light blue droplets and aligning cell walls in the phloem and vessel elements. (**b**) Larger magnification of the cells (mainly vessel elements) and areas of the perimedullary region, note the tylosis-like structures (arrowhead) and suberin-coating in vessels. Perimedullary parenchyma cells and axial parenchyma aligning vessels are filled with phenolic aggregates and droplets, surrounding the ground tissue of the pith. (**c**) Grainy deposits. (**d**) Characteristic yellow auto-fluorescence of polyphenolic compounds, which were especially pronounced in the axial parenchyma but also in some vessel elements. Hyphae indicated by white arrowheads. (**e**) Phenolic compounds accumulating in sclerenchyma, axial parenchyma, and vessel elements, together with fungal hyphae (black arrowheads). (**f**) Staining with Nile red shows neutral lipids as yellow–gold droplets when illuminated with UV-light (330–380 nm) in the rachis section from infected clone 27x. C: cortex; Ph: phloem; Xy: xylem; P: pith; Sc: sclerenchyma; Pm: perimedullary parenchyma; Ve: vessel elements; Ap: axial parenchyma; Pp: polyphenolics. Scale bars: (**a**), 100 µm; (**b**–**d**,**f**), 50 µm; (**e**), 20 µm.

**Table 1 microorganisms-10-00375-t001:** Lesion size and level of *Hymenoscyphus fraxineus* DNA in inoculated rachises of common ash (four clones) and Manchurian ash (one clone) 16 weeks after inoculation. Sampling points are as follows: S1, at the border of the necrosis; S2, 1 cm away from the first sample towards the petiole base; S3, petiole base; and S4, petiole–twig junction. Samples in bold were examined with light microscopy.

Common Ash	Susceptibility Classification in Field	Lesion Length (mm)	Necrosis Circumference (%)	*H. fraxineus* DNA pg/mg Tissue			
				S1	S2	S3	S4
35 (*n* = 6)	Less susceptible	10	50	13.63	0	0	0
		7	50	22.50	0	0.65	0
		**8**	**80**	**17.79**	**4.62**	**0.38**	**0.11**
		12	100	142.44	1.20	0.07	0.08
		14	80	319.22	0	0	0
		10	100	31.45	1.57	0	0.12
14165 (*n* = 7)	Less susceptible	16	30	0.94	0	0.03	0.06
		**4**	**50**	**146.09**	**0.21**	**0.01**	**0.03**
		13	80	0.27	0	0	0.09
		14	70	108.27	7.93	0	NA
		8	70	519.83	17.36	0	NA
		5	95	211.81	12.21	0.05	NA
		8	60	129.36	9.57	0	NA
40 (*n* = 7)	Susceptible	12	100	171.06	4.74	0.35	NA
		13	100	182.85	0.34	0	NA
		6	30	433.43	0	0	0
		17	100	0.27	0.61	0.11	0
		27	40	0	0.37	0	0
		18	100	2.9	0	0	NA
		9	80	7.03	0	0.14	0.14
27x (*n* = 5)	Susceptible	15	100	0	0	1.27	0
		38	100	540.26	83.40	0.47	0
		16	80	3223.64	99.41	0.39	NA
		20	100	13.23	0	0.03	NA
		**10**	**100**	**1546.03**	**2627.45**	**597.47**	**0.14**
Manchurian ash	Less susceptible	**15**	**50**	**8881.49**	**3835.71**	**207.55**	**2.27**
1989-0095-4		12	90	383.45	674.00	6.64	NA
(*n* = 3)		10	100	250.46	19.72	NA	NA

**Table 2 microorganisms-10-00375-t002:** Summary of the degree of necrotic tissue, level of pathogen DNA, and histological observations of hyphal abundance and prevalence of starch from samples of four clones subjected to rachis-inoculation by *Hymenoscyphus fraxineus* and one clone inoculated with sterile plugs (Control). Ash dieback susceptibility in field: less susceptible; common ash clones 35, 14165 and Manchurian ash clone 1989-0095-4 and susceptible; common ash clone 27x.

						Phloem		Axial Parenchyma		Perimedullary Parenchyma	
Clone	Treatment	Lesion Length (mm)	Necr. Circumf. (%)	Sample Site ^1^	DNA (pg) ^2^	Hyphae	Starch	Hyphae	Starch	Hyphae	Starch
35	Cont.	0	0	Lesion border	0	0	7	0	10	0	5
				1 cm from border	0	0	10	0	10	0	10
				Petiole base	0	0	10	-	-	-	-
				Petiole/twig junc.	0	0	10	-	-	-	-
35	Inoc.	8	80	Lesion border	17.8	3	7	1	7	0	9
				1 cm from border	4.6	2	8	1	8	2	6
				Petiole base	0.4	0	9	-	-	-	-
				Petiole/twig junc.	0.1	0	10	-	-	-	-
14165	Inoc.	4	50	Lesion border	146	0	10	0	10	2	10
				1 cm from border	0.2	0	10	0	10	0	10
				Petiole base	0.006	0	10	-	-	-	-
				Petiole/twig junc.	0.03	0	10	-	-	-	-
27x	Inoc.	10	100	Lesion border	1546	6	0	8	0	10	0
				1 cm from border	2628	5	0	8	0	10	10
				Petiole base	598	9	1	-	-	-	-
				Petiole/twig junc.	0.1	0	10	-	-	-	-
1989-	Inoc.	15	50	Lesion border	8882	10	1	10	0	10	0
0095				1 cm from border	3836	10	1	9	0	10	1
-4				2 cm from border	3740	10	1	10	2	10	0
				Petiole base	208	0	10	-	-	-	-
				Petiole/twig junc.	2.3	0	10	-	-	-	-

Samples were collected 24 September 2018, except for 1980-0095-4, of which the leaf dropped off on 4 September 2018. ^1^ Samples were collected from the lesion border and towards the stem, as illustrated in Figure 1a, ^2^
*H. fraxineus* DNA pg/mg ash tissue. Microscopy observations of hyphae and starch grains were based on the examination of 10 randomly chosen microscopic fields with the objective lens of 100× magnification (0.0314 mm^2^); a value of 10 indicates that the feature was present in all 10 microscopic fields, while a value of 0 means that the feature was not observed in any of the 10 microscopic fields. The missing values for axial parenchyma and perimedullary parenchyma were due to these cell types being not detected in the samples from the petiole base and petiole/twig junction.

## Data Availability

Data is contained within the article or Appendix A.

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
