# Peer review of "Host–Pathogen Interactions in Leaf Petioles of Common Ash and Manchurian Ash Infected with Hymenoscyphus fraxineus"

_microorganisms, 2022, doi:10.3390/microorganisms10020375_

Round 1

Reviewer 1 Report

This article deals with the differences between Fraxinus clones in tissue colonization by the pathogen Hymenoscyphus fraxineus and host defence responses. Although the study is based on a relatively small number of investigated plants as well as a small dataset for statistical analyses, the results obtained are of great importance for understanding the relationship between host and fungal pathogen.

The manuscript is well written and interesting to read. Some minor corrections are needed:

Authorities for the hosts and pathogens should be included at first mention of the taxa.

The 18% PDS for clone 27x in Supplemental Table 1 does not match the 17.5% in the text (lines 82 and 201).

Line 92: Is the diameter of ash wood plugs used for inoculations correct? 0.2 mm in diameter… Didn't you mean 0.2 cm?

The data on the number of inoculated ramets / leaves in Section 2.2 do not correspond to the numbers (n) in Table 1 as well as in the legend of Figure 2.  See section 2.2 (line 95): “Three ramets of each clone were inoculated on two or three compound leaves.” .... Table 1: 27x (n=5); Manchurian ash n=3 .... Figure 2: “…. two inoculated leaves from two ramets of 1989-0095-4.”  

Suggest adding n measurements per each clone in () after clone names in Figures 2a and 2b.

Figure 3: add “T: Tylosis” to the Explanations.

Author Response

Reviewer 1

Reviewer comment

Author response

Authorities for the hosts and pathogens should be included at first mention of the taxa

We have now included the authorities for hosts and pathogens at mention in text.

The 18% PDS for clone 27x in Supplemental Table 1 does not match the 17.5% in the text (lines 82 and 201).

We have changed “17.5%” in the text to 18%

Line 92: Is the diameter of ash wood plugs used for inoculations correct? 0.2 mm in diameter… Didn't you mean 0.2 cm?

Yes, thank you, we mean 0.2 cm and have now corrected this in the text.

The data on the number of inoculated ramets / leaves in Section 2.2 do not correspond to the numbers (n) in Table 1 as well as in the legend of Figure 2.  See section 2.2 (line 95): “Three ramets of each clone were inoculated on two or three compound leaves.” .... Table 1: 27x (n=5); Manchurian ash n=3 .... Figure 2: “…. two inoculated leaves from two ramets of 1989-0095-4.”

We agree that is was not clear why the numbers did not correspond to each other. The reason for the lower final sample size of 27x and Manchurian ash relative to the number of performed inoculations is that some of the rachises unfortunately were lost in the greenhouse during the experiment. We have tried to clarify this in the text.

Suggest adding n measurements per each clone in () after clone names in Figures 2a and 2b

We decided to add the sample size per clone in the text, as we were afraid that the Figures would be too crowded and also because all four diagrams (a, b, c, d) are based on the same number of inoculated rachises.

Figure 3: add “T: Tylosis” to the Explanations.

We have added “T” for tylosis as suggested.

Reviewer 2 Report

Generally, the experimental design is easy to understand but at the end the data described and discussed unclear.

Despite the authors represented a hypothesis to confirm in this study, the overall aim of the research seems was not stressed. Moreover, both in abstract and in conclusion there is not information on this hypothesis, how the data obtained will be useful for development of ash varieties resistant to dieback, what particular disease characteristics or histochemical features will be useful as the selection markers.

Tables 1 and 4 should be simplified for understanding: less abbreviations, less size if possible.

Tables 2 and 3 seems can be transformed to text showing just statistically significant data and high level correlations.

Histochemical study on resistance mechanisms seems are not included in the research tasks. It looks as a different job.

Author Response

Reviewer 2:

Reviewer comment

Author response

Despite the authors represented a hypothesis to confirm in this study, the overall aim of the research seems was not stressed.

Moreover, both in abstract and in conclusion there is not information on this hypothesis, how the data obtained will be useful for development of ash varieties resistant to dieback, what particular disease characteristics or histochemical features will be useful as the selection markers.

We have tried to meet the critiscism given by the reviewer by making the following changes:

We have modified the title so that it better reflects the content of the research.  

We have also added the following conclusion in the manuscript: Our results support the hypothesis that there is a connection between leaf and shoot susceptibility as reduced spread of the pathogen was found in clones with relatively strong field performance. However, we found no relation between degree of petiole colonization and histochemical features related to defence responses.

Considering the small number of clones used, however, we do not feel that our results can yet be used for future breeding of resistant ashes and prefer not to include speculations on this aspect in the manuscript. Instead, we have added suggestions for future research that can bring research forward on ash and ash dieback interactions and thus pave the way towards a well-founded breeding program.

Tables 1 and 4 should be simplified for understanding: less abbreviations, less size if possible.

We have simplified Table 1 and 4 (now called Table 1 and 2) by using fewer abbreviations, and for Table 1, we also removed the simple averages and standard deviations.

Tables 2 and 3 seems can be transformed to text showing just statistically significant data and high level correlations.

We have followed the reviewer’s suggestion by including statistically significant data and high-level correlations in the text, and by removing the Tables 2 and 3.

Histochemical study on resistance mechanisms seems are not included in the research tasks. It looks as a different job

Our second hypothesis concerns histochemical analysis but our results do not support the hypothesis. As mentioned above, we have specified this in our conclusion.

Moderate English changes required

A native English speaker has read and improved the language.